# Effects of Soil pH and Fertilizers on Haskap (*Lonicera caerulea* L.) Vegetative Growth

Catherine Tremblay [1], Annie Deslauriers [1] , Jean Lafond [2], Julie Lajeunesse [2] and Maxime C. Paré [1,*]

[1] Laboratoire d'écologie végétale et animale, Département des sciences fondamentales, Université du Québec à Chicoutimi, Saguenay, QC G7H 2B1, Canada; catherinetremblay77@gmail.com (C.T.); Annie_Deslauriers@uqac.ca (A.D.)

[2] Agriculture and Agri-Food Canada, Quebec Research and Development Centre, Normandin Research Farm, Normandin, QC G8M 4K3, Canada; lafondj@agr.gc.ca (J.L.); julie.lajeunesse@canada.ca (J.L.)

* Correspondence: maxime.pare@uqac.ca; Tel.: +1-(418)-545-5011 (ext. 5071)

**Abstract:** Haskap (*Lonicera caerulea* L.) is a new northern latitude fruit crop that is increasing in popularity. This sudden enthusiasm for haskap increases the need for obtaining baseline knowledge related to establishing it as a crop, such as its optimal soil pH and fertilizer needs. In a greenhouse, one-year-old haskap plants (cultivar: Indigo Treat©) were grown in a local loamy sand. We assessed the impact of pH and fertilizer on haskap vegetative growth through an experiment involving four soil pH and five fertilization treatments of three N sources (ammonium, nitrate, and organic (chicken manure)). Leaf senescence as well as above-ground and root biomass were recorded after 19 weeks of vegetative growth. For cultivar Indigo Treat©, optimal vegetative growth was observed under slightly acidic soil conditions (pH$_{CaCl2}$ 5.5–6 or pH$_{water}$ 5.9–6.5) without application of N. Phosphorus and K fertilizers did not influence vegetative growth. We here discuss the implications for establishing haskap orchards.

**Keywords:** blue honeysuckle; honeyberry; nitrogen; northern agriculture; small fruit

## 1. Introduction

Haskap is a new fruit-producing crop that is growing in popularity in northern regions of America and Europe. Haskap has a number of phytochemical and antioxidant properties, which may improve human health by reducing blood glucose and cholesterol levels [1,2]. Although haskap is originally native to several parts of the world (northern regions of America, Asia, and Europe) [3–5], the development of new cultivars by the University of Saskatchewan Fruit Program (Canada) has recently boosted its popularity. Today, haskap orchards are already well established in America (United States and Canada) and worldwide (Russia, Japan, Czech Republic, Poland, and China) [6]. According to Quebec's haskap association (Canada), there are about 730,000 individual haskap plants (~350 ha) growing throughout the province of Quebec and half of these plants are located in the Saguenay-Lac-Saint-Jean region of the province [7]. This sudden and significant enthusiasm for haskap demands the development of baseline knowledge in regard to suitable soil conditions for its cultivation, as presently very little information has been published for this crop [8].

Correctly managing and preparing soil prior to establishing an orchard is critical to maximize plant establishment, vegetative growth (both above- and below-ground), and plant health. From a producer's perspective, favoring plant establishment minimizes costs during the initial four- to five-year period of high investment and low income, which normally lasts from the first planting to the first significant harvest [9]. Other than managing weeds, adjusting soil pH and applying fertilizers

are the main management practices that are performed by producers before and during the years of orchard establishment. Currently, soil pH and fertilizer requirements for the optimal growth of haskap are inferred based on the optimal conditions for other small fruits, such as raspberry and highbush blueberry [9]. Therefore, it is suggested to establish haskap orchards on slightly acidic soils and to fertilize the crop annually with significant amounts of nitrogen (N), phosphorus (P), and potassium (K) [9]. On the other hand, multiple cultivars (and thus a wide genetic diversity) are used for breeding haskap [10,11], making it hard to predict the forms of N (i.e., N-organic, N-ammonium, or N-nitrate) that haskap most prefers; the functional traits and provenance of a plant dictate its ability to use one form of N over another [12].

Here, we investigate the effect of soil pH and N fertilizers on the second-year vegetative growth of haskap in a greenhouse setting. We hypothesize that haskap growth will be highest under slightly acidic soil conditions and with N supplied as calcium nitrate.

## 2. Materials and Methods

This study was carried out in a greenhouse at the Université du Québec à Chicoutimi during the winter of 2016. Plastic growing pots (4.5 L) were filled gently with 3.5 kg of Mistassini loamy sand (air-dry soil basis) [13] provided by Les Camerises du Lac Inc., Labrecque, QC, Canada (see the reported chemical properties of the soil in Table 1). The collected topsoil (top 0–15 cm of the soil profile) was sieved at 10 mm and then thoroughly homogenized on a tarp with a rake before potting.

**Table 1.** Initial chemical properties of the mineral soil (Mistassini loamy sand) used for this study.

| Soil Properties | Method | Value |
| --- | --- | --- |
| Soil pH | In water (1:1) | 6.4 |
| Soil organic matter (%) | Combustion | 5.4 |
| Total nitrogen ($g \cdot kg^{-1}$) | Kjeldahl | 1.4 |
| P ($mg \cdot kg^{-1}$) | Mehlich 3-Extractable | 5.4 |
| K ($mg \cdot kg^{-1}$) | Mehlich 3-Extractable | 53.1 |
| Mg ($mg \cdot kg^{-1}$) | Mehlich 3-Extractable | 17.3 |
| Ca ($mg \cdot kg^{-1}$) | Mehlich 3-Extractable | 1284 |
| Al ($mg \cdot kg^{-1}$) | Mehlich 3-Extractable | 2097 |
| Fe ($mg \cdot kg^{-1}$) | Mehlich 3-Extractable | 67.8 |

We tested four soil pH treatments (Table 2). Lime (92% $CaCO_3$ and 2% $MgCO_3$, Graymont, Saint-Marc-des-Carrières, QC, Canada) was used to increase soil pH, whereas an acidic solution (0.4 M HCl) was applied to decrease soil pH [14]. The soil samples were kept moist for 11 weeks to equilibrate and homogenize the soil pH within each pot. Soil pH was monitored weekly during this initial phase and then bimonthly during the experimental period. Soil was collected for pH analysis using an extractor tray (open rod) of an increment borer (normally used for tree-ring sampling). The extractor tray allowed us to sample the entire soil profile (15 cm deep) with minimal disturbance (~0.5 cm diameter holes). Soil pH was then read using a pH meter (AR25 pH Meter, Fisher Scientific, Hampton, New Hampshire, USA) in the soil and in a 0.01 M $CaCl_2$ solution (1:2 ratio) [15]. Compared to measurements of soil pH in water, measurements in a $CaCl_2$ solution are more consistent and representative [16,17].

We applied five fertilization treatments to each of the four soil pH treatments as a secondary factor (Table 2). We selected granular forms of P, K, and organic (granulated poultry manure, Acti-Sol©, 5-3-2, C/N-ratio of 10) fertilizers that we mixed thoroughly into the soil. In each pot, we planted a single one-year-old haskap seedling in dormancy (18 ± 4 cm in height, Indigo Treat©, Végétolab, Alma, QC, Canada). We used physiological attributes (e.g., length of the stems) to select the most similar and representative haskap plants. Before planting, the roots were soaked twice in deionized water, 10 min for each soaking, to remove any residual fertilizers that may have remained within the rooting zone. Mineral N fertilizers (ammonium sulfate and calcium nitrate) were dissolved into water used

for irrigation. However, unlike P, K, and the organic fertilizers, mineral N fertilizers were split into four applications at one-month intervals (1 g of N per application per month per pot). The N solutions were applied around the plant collet to minimize direct contact between the solution and the roots. Mistassini loamy sand allowed for the immediate and adequate distribution of the solution throughout the soil.

**Table 2.** Experimental factors used for this study.

| Factor | pH Category | Soil pH Class [1] | $pH_{CaCl2}$ | $pH_{water}$ [1] | |
|---|---|---|---|---|---|
| Primary factor | pH1 | Strongly acidic | 4.3–4.6 | 4.7–5.0 | |
| | pH2 | Moderately acidic | 5.1–5.4 | 5.5–5.8 | |
| | pH3 | Slightly acidic | 5.5–6 | 5.9–6.5 | |
| | pH4 | Neutral | 6.3–7 | 6.8–7.5 | |
| | **Fertilizer type** | **N–P–K** [2] | **N** | **P** | **K** |
| | | g plant$^{-1}$ | | Source | |
| | C | 0–0–0 | — | — | — |
| | T | 0–2.9–1.7 | — | $Ca(H_2PO_4)_2$ | KCl |
| Secondary factor | M1 | 4–2.9–1.7 | $(NH_4)_2SO_4$ | $Ca(H_2PO_4)_2$ | KCl |
| | M2 | 4–2.9–1.7 | $Ca(NO_3)_2$ | $Ca(H_2PO_4)_2$ | KCl |
| | O | 4–2.9–1.7 | Granulated poultry manure | | |

[1] Soil pH class and soil $pH_{water}$ obtained from Vanasse et al. [18]. [2] Fertilizers normalized on equivalent basis ($N_{total}$, $P_2O_5$, and $K_2O$).

To reflect early summer conditions in the Saguenay-Lac-Saint-Jean region, greenhouse air temperatures were set at 18 °C during the day and 11 °C at night; the daily photoperiod was fixed at 14 h. The plants were irrigated manually to keep the soil water content between 0.15 and 0.20 m$^3$·m$^{-3}$, representing about 50–80% of the soil field capacity. A time-domain reflectometer probe (GS3, Decagon Devices, Pullman, Washington, United States) combined with a portable data logger (Procheck, Decagon Devices, Pullman, Washington, United States) was used weekly to verify and, if needed, adjust the soil moisture in each pot. Water was gently added to avoid nutrient leaching from the bottom of the pots. The pH of the water used for irrigation was adjusted throughout the experiment to maintain the pH of the treatments.

Plants were harvested 19 weeks after planting. Final (at harvest) dry above-ground (leaf and stem) and root biomass (g) were determined after drying the samples at 65 °C for 24 h. Prior to drying, roots were gently washed in water to remove soil particles. Ten seedlings were used to estimate initial (before planting) above- (mean 0.41 ± 0.08 g) and below-ground biomass (mean 0.43 ± 0.10 g). Leaf nitrogen concentrations were indirectly estimated using a chlorophyll meter (SPAD-502, Konica Minolta, Tokyo, Japan) two (week #15) and four (week #17) weeks after the last mineral N application. A combined average of four non-senesced leaves per plant was used for SPAD analysis. In addition, the percentage of leaf senescence (%) was calculated as the weight of the leaves that fell (senescence) prior to the end of the experiment (g), divided by the final total leaf biomass (g), multiplied by 100. Finally, a plant was considered as defoliated when four leaves out of five were senesced.

All treatments (20 treatments: 4 soil pH × 5 fertilizer) were replicated five times (100 experimental units) in a split-plot experimental design (to facilitate irrigation), where soil pH and fertilization served as primary and secondary factors, respectively (Table 2). We ran analysis of variance (ANOVA) on the results using the lmerTest package in R [19,20]. When results from ANOVA were significant, Tukey post-hoc tests determined the treatments having significant differences. The applied ANOVA model differed for soils fertilized with ammonium sulfate (M1) and granulated poultry manure (O) because of difficulties in adequately controlling soil pH after the application of these forms of fertilizer. To address this issue for these treatments, soil pH (median = 5.62) was used as a covariate. We used the car package in R to perform analysis of covariance (ANCOVA) [21]. When necessary, response variables were transformed (e.g., logarithm (log)) to ensure a normal distribution of the data.

## 3. Results

Leaf senescence and both above-ground and root biomass were optimal (lowest senescence and highest biomass) at slightly acidic conditions of treatment pH3 ($pH_{CaCl2}$ 5.5–6) (Figure 1). Compared to the control, adding mineral N as calcium nitrate (M2) did not significantly influence chlorophyll meter values (week #15: $p = 0.352$; week #17: $p = 0.421$ (results not shown)) but it markedly reduced root biomass by about 50% only for treatments pH3 ($pH_{CaCl2}$ 5.5–6) and pH4 ($pH_{CaCl2}$ 6.3–7) (Figure 1). Compared to the control (C), adding P + K (T) did not influence any of the measured variables (Figure 1). Relative to ammonium sulfate (M1), adding organic fertilizer (O) did not affect significantly leaf senescence and root biomass; however, it reduced above-ground biomass by about 40% (Figure 1). On average, half of the haskap plants were considered as defoliated after nine and 11 weeks for pH1 and pH2, respectively, whereas less than 20% of haskap plants were defoliated at the end of the experiment (week #19) for pH3 and pH4 (results not shown). Similarly, half of the haskap plants were defoliated at week #10 and #14 for (M1 and O) and M2 fertilizer treatments, respectively, whereas less than 15% of haskap plants were defoliated at the end of experiment for C and T treatments (results not shown).

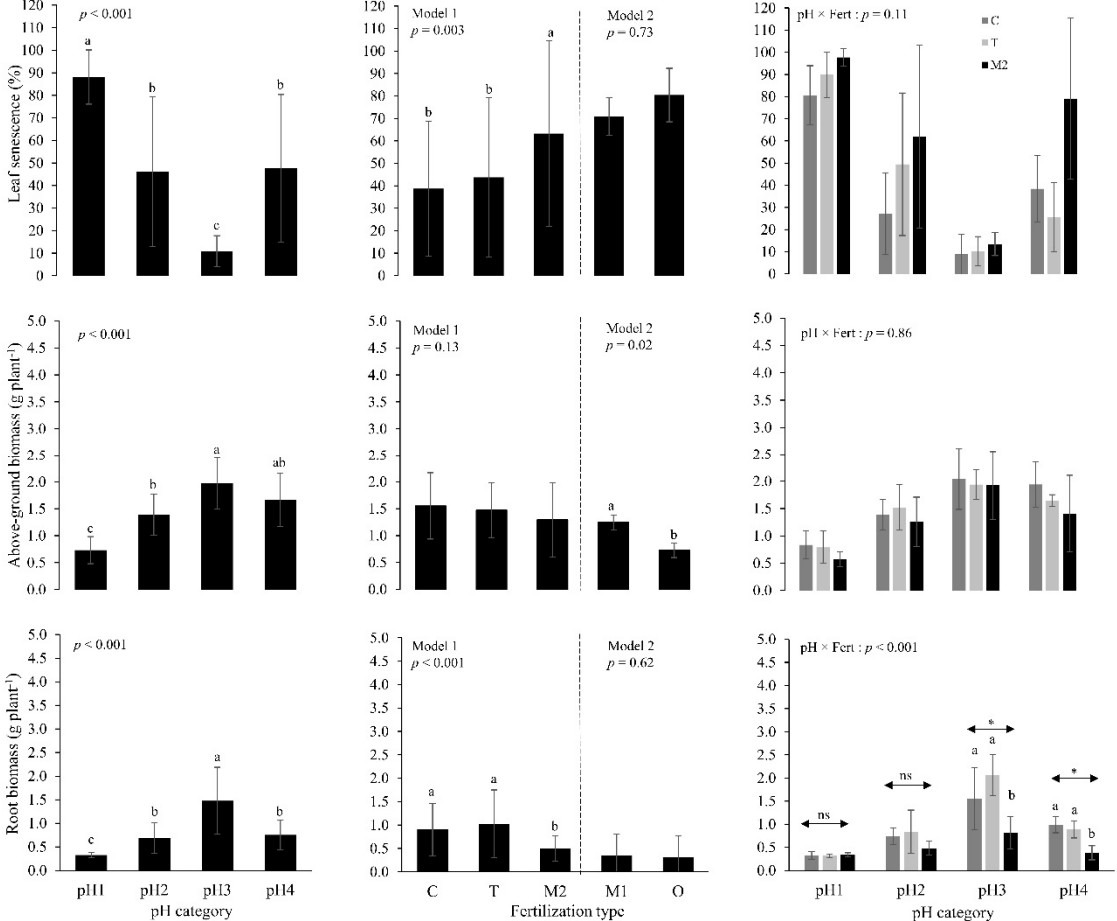

**Figure 1.** Leaf senescence, above-ground and root dry biomass at harvesting, after 19 weeks of vegetative growth in a greenhouse, for treatments with differing soil pH levels and types of fertilizer applied. Error bars represent the standard deviation from the mean. Letters indicate significant differences between the combined averages (main effect figures) and average (interaction figures) based on Tukey's post-hoc tests. pH1 = strongly acidic; pH2 = moderately acidic; pH3 = slightly acidic; pH4 = neutral (see Table 2 for the pH ranges of each category); C = without fertilizer; T = P + K (without N); M1 = N + P + K, N as ammonium sulfate; M2 = N + P + K, N as calcium nitrate; O = N + P + K, as organic fertilizer (granulated poultry manure).

## 4. Discussion

Optimal vegetative growth for haskap under greenhouse conditions occurred in slightly acidic soil conditions; thus, haskap had maximum growth in the pH range of most cropped species [18,22]. Above and below these values ($pH_{water}$ 5.9–6.5), root development was markedly compromised and leaf aging was greater, therefore likely reducing long-term vegetative growth. A reduction in root biomass at lower soil pH values has been reported for many domesticated and wild plants, as root development can be compromised by increased aluminum availability in soils having a lower pH [23–27]. On the other hand, alkaline soil conditions can also reduce root biomass, since an increase in calcium activity within the soil solution decreases the ability of the plant to uptake other nutrients, such as phosphorus and manganese [28]. Increased leaf senescence for treatments pH1 and pH2 may also be due to the chlorine added through the HCl used for acidifying the soil, as leaves of woody plants (e.g., trees, vines, shrubs) are generally sensitive to chlorine concentrations in soils [29]. However, the sensitivity of haskap to chlorine concentrations in soil remains to be demonstrated. Opposite trends in root biomass and leaf senescence (Figure 1) also suggest that leaf aging was likely a result of the inability of the plant to maintain a large leaf area with a limited root biomass. Plants must balance their allocations between leaves and roots so as to match the physiological activities and functions performed by these organs [30].

Although the soil used was initially very nutrient poor (Table 1), adding P + K fertilizers did not improve haskap vegetative growth. Moreover, adding N fertilizers (regardless of the N source) likely increased stress by strongly reducing root development and increasing leaf senescence and defoliated plants (Figure 1). The negative impact of N fertilizers on root development has been documented for other crops, and two mechanisms may explain this pattern. First, stress from increased soil salinity caused by fertilizer applications may decrease root development [31]. However, the K fertilizer used in our study (KCl) is also known to increase markedly soil salinity [32], whereas its use had an insignificant impact on root development and leaf senescence in our treatments (Figure 1). Second, plants may inversely adapt their root development to soil concentrations of N [33,34]; plants invest less energy in root development where concentrations of soil N are high, as reflected by our lower root biomass and similar chlorophyll meter values for all N fertilizers, regardless of the N source or type (Figure 1). Although both mechanisms may occur simultaneously, we believe the second mechanism (i.e., a lower investment in root growth with use of N fertilizers) better explains our results.

Many haskap producers fertilize their orchards during the establishment period using significant amounts of N (2–4 g of N $plant^{-1}\cdot year^{-1}$) around the plant collet and under an impermeable plastic mulch (polyethylene) [9]. As haskap root growth should be favored and leaf aging (senescence) should be minimized (to expand the volume of soil and above-ground areas that can be exploited), our results suggest that producers should not use N fertilizers, at least during the establishment period. Phosphorus and K fertilizers might be used (if judged as necessary), as they did not influence significantly root and overall vegetative growth.

Finally, it remains possible that our results are specific to only a few haskap cultivars (Indigo Treat© was used in this study), as a large genetic diversity exists among phenotypes [8]. Furthermore, results should be interpreted cautiously; mineral N concentrations were not directly monitored throughout this experiment. To assure transferability to field conditions, a larger study involving multiple cultivars of haskap over a longer period of time and under various field conditions would be ideal in developing robust recommendations for farmers and stakeholders.

**Author Contributions:** The paper is the result of the collaboration among all authors. Conceptualization, M.C.P. and C.T.; methodology, C.T. and M.C.P.; validation, M.C.P., A.D., J.L. (Jean Lafond), and J.L. (Julie Lajeunesse); formal analysis, C.T.; investigation, C.T.; resources, M.C.P.; writing—original draft preparation, C.T.; writing—review and editing, MC.P., A.D., J.L. (Jean Lafond), and J.L. (Julie Lajeunesse); visualization, C.T.; supervision, M.C.P. and A.D.; project administration, M.C.P.; funding acquisition, M.C.P.

**Funding:** This study was funded by the Fonds de développement de l'Université du Québec à Chicoutimi (FUQAC), the Fonds de recherche axé sur l'agriculture nordique (FRAN-02), and the Fonds de recherche du Québec—Nature et technologies (FRQNT).

**Acknowledgments:** The authors thank Végétolab Inc. and Les Camerises du Lac, Inc. for providing plants and soils, respectively.

**Conflicts of Interest:** The authors declare no conflict of interest. The funders had no role in the design of the study; in the collection, analyses, or interpretation of data; in the writing of the manuscript, or in the decision to publish the results.

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
