# Peer review of "Effects of Soil pH and Fertilizers on Haskap (Lonicera caerulea L.) Vegetative Growth"

_agriculture, doi:10.3390/agriculture9030056_

Reviewer 1 Report

Haskap is indeed a rapidly emerging crop and growers are investing significant funds in orchard establishment in anticipation of a profitable return 5+ years later. There is very little guidance relating to soil fertility management for this crop, so indeed this sort of research is novel and timely.

I found the experiment to be quite limited in its design and reliability. I would prefer to have seen stronger replication in the form of repeating the trial, second variety, and/or second soil type. The fertilizer treatments were not adequately described, a rationale for them was not clearly provided, and were limited to essentially one rate albeit testing for multiple nutrients. If the goal was to compare fertilizer types, then this should have been clearly described and hypothesized. Overall, the data, especially considering it was a greenhouse trial, seemed like a preliminary dataset and would lay the groundwork for more thorough study. While senescence was

Title: Based on the research presented, I do not think the title should indicate “Optimal…. fertilizer application…”. This is deceiving. Multiple fertilizer application rates were not tested (only one rate was compared with the control), the application rates were very low, and their was no assessment of nutrient uptake by the plant.

27 America or “North America”? also grown commercially in Northern Europe

30 Is “tastier” really the key criteria for selection in this breeding program? Please revise.

35 “obliges”?

40 not clear how favouring plant establishment minimizes costs

53 does “optimal” mean highest?

58 is it necessary to provide the name of company providing pots?

59 & Table 1 – you must clearly indicate whether the soil was tested in-house or at a soil testing lab, and in either case the extraction technique used to determine the soil nutrient status. Also, soil test reports typically present P as P2O5 and K as K2O. If you are being intentional about the use of actual “P” and “K” then the table is fine, but I will note that it does not reflect normal soil test reports (that I am familiar with).

74 What is the analysis of the Acti-Sol fertilizer? On the website it is reported as 5-3-2. In poultry manure pellets, the N analysis includes organic N which is not all available in the first growing season. Did you account for N availability?

75 What was the condition of the seedling when planting? Was it already leafed out or just a stem?

81-82 You only have one fertility rate of 4 g N per plant (compared with the control). How did you decide this this was the optimum? If we assume that there are 2500 plants/ha, this amounts to an application rate of 10 kg/ha of N being applied. For the poultry manure, this means likely 5 kg/ha of available N. These are very low application rates, and likely do not exceed background levels of N. Similarly, the levels of P and K applied seem low.

85. In Table. 2 are you really applying N-P-K or N-P2O5-K2O? It looks like you probably standardized your chemical fertilization rates to match the proportions provided in the Acti-Sol. If this is correct, then you should state this. And if this is the case, then I assume you added 100g of Acti-Sol dry weight to each pot.

85 Table 2 also – What does “Fertilizer Type” mean? The C, T, M1, M2 and O need to described in the text when you list your fertilizer treatments and at the bottom of the table.

98-99 Root and shoot biomass was assessed before planting at around 0.4 g each (0.84g/plant total). Was this included or deducted from the data in Figure 1? I am guessing it is included and that these figures are provided for reference. However, you could have used this to help describe the growth of the plants.

115 & 123 Again the word optimal. What does optimal mean/represent? Do you mean lowest? Highest?

117 reduce by about 50%, I assume compared with the Control?

156-7 I find the recommendation relating to P and K to be very generic. Only one rate of these nutrients was applied in small amounts and to only one soil type.  

A key limitation of this study was that only one soil type was used.

Is soil pH was monitored throughout the experiment, perhaps it would be useful to show the data?

Was electrical conductivity measured in the soil before and after treatments were applied and stabilized?

The fact that M1 and O could not be compared with a Control is a real shortcoming of the experiment

Have you considered ammonia/ammonium toxicity as a potential problem for M1 and O? See for example: https://dl.sciencesocieties.org/publications/aj/articles/108/6/2485

Leaf senescence was highlighted as an important measure. However, it is not clear if the mass of senesced leaves is included in the biomass measurement. It would also be good to know when the leaves actually dropped. The experiment was carried out for 19 weeks, which would account for most of a growing season. Was the leaf senescence all towards the end of the 19 weeks or did it come earlier which would indicate stress?

In Figure 1 interaction for senescence the standard deviation varied tremendously across treatments suggesting that there was a lot of variability in the plant material or the treatments. More replicates may have been needed, or perhaps there were outliers?

There was no indication of plant health provided. It may have been difficult to assess leaf tissue nutrient status with the limited leaf volume available, but were there any symptoms of nutrient deficiency? Leaf chlorophyll could have been assessed with a simple SPAD meter.

Unlike the promise in the abstract, there was very little discussion of management implications. The pH assessment was the strongest data, but there was no discussion about options for soil pH modification in a real field setting.

147 Comments regarding plant response to N were poorly referenced. The only reference provided was relating to Arabidopsis. There could be significantly more discussion around N dynamics in the soil and plant growth, especially in relation to the form of fertilizer.

P & K are very important for root establishment this could be discussed.

Author Response

See the point-by-point responses (reviewer #1) in the attached file.

Reviewer 2 Report

Dear author and co-authors,

Overall, this is a clear and well-written manuscript for a short communication. Since haskap attracts attention of fruit growers not only in Canada, the research topic is definitively of recent relevance. A pot experiment approach is generally appropriate, although clarification of a few details should be provided, particularly to assure transferability of the results to field conditions.  Please find specific minor and major comments in the following

 Minor comments:

 (1)    L 33: Giving the growing area instead of the number of plants would provide a better impression of the current relevance and development of haskap production compared to other crops.

 (2)    L 48: Please correct „cutlivars“

 (3)    L 57: 3.5 kg of field-fresh soil or 3.5 kg soil dry matter?

 (4)    L 61 (Tab. 1): Please give brief information on the soil analysis methods used. A P soil content of 5.4 mg kg-1 would be very low when using the CAL or the DL-method.

 (5)    L 74/Tab. 2:

5a) Are „Acti‐Sol©“ (L74) and „granulated poultry manure“ (Tab. 2) the same?

5b) Please give more detailed information on the organic fertilizer (N-, P- and K-content, C/N-ratio)

5c) The organic fertilizer probably contained significant amounts of nutrients others than N (particularly P). How was the same P and K fertilization rate as of the mineral fertilizer treatments realized? Please add this information to the materials and methods chapter.

 (6)    L 94/95: Please give additional information on how and to which pH-levels the irrigation water was adjusted?

 (7)    Fig. 1:

7a) Does “combined averages” mean, that mean values of the respective other factor are shown? This would not be true for the right figure column. Please describe more precisely.

7b) Please specify the units given on ordinate axis: e.g. “g plant-1”?

   Major comments:

(1)    L 108-111/Fig. 1: Applying ANCOVA to statistically correct the results for the (actually undesired) fertilizer effect on pH is generally appropriate. However, to my understanding there is no need to separate the treatments into two groups (Model 1: C, T, M2; Model 2: M1, O). If the ANCOVA model would be applied to entire data sets (C, T, M1, M2, O), also M1 and O could be compared to the control treatment C.

 (2)    Tab. 2: In order to assure transferability of pot experimental results to field conditions, please explain how the chosen fertilizer levels of 4.0 (N), 2.9 (P) and 1.8 (K) g pot-1 were set. Adding 4 times 1 g of mineral N to 3500 g soil results in an accumulated soil mineral N concentration of approximately 1300 mg N kg-1 dry soil (assuming 3.5 kg of field-fresh soil with a gravimetric water content of 12% and ignoring plant N uptake), which is extremely high and can be assumed to cause salt stress effects. Due to initial soil mineral N at planting (please add value to table 1) and additional N released from soil organic matter during the experiment, soil mineral N concentration at harvest might have even been higher. Since practical orchard fertilizer rates of 2 to 4 g plant-1 will be distributed to a much higher soil volume (by larger application area and deeper soil incorporation/leaching), the transferability of the obtained results to field conditions is questionable. This should, at least, be discussed critically.

 (3)    L 152-156: Please consider major comment 2 when drawing conclusions from the obtained N fertilizer effects. Furthermore, please also consider that only selected vegetative growth parameters and no effects of N fertilization on yield formation were investigated.

 (4)    Please discuss also the negative effect of the organic fertilizer (O) on above-ground biomass when compared to M1. Plant N availability of organic fertilizers strongly depend on their C:N ratio (or Nt-content) and is generally lower than of mineral N fertilizers. The negative effects of O can therefore not be explained by higher soil mineral N concentrations – at least without soil mineral N sampling results.

Author Response

See the point-by-point responses (reviewer #2) in the attached file.

Round  2

Reviewer 1 Report

Thank you for the clarifications in the text and the responses that you provided to my comments.

You used leaf senescence as a key indicator of treatment performance, but aside from measuring this (L102) there is no further description. I do still think that a bit more clarification and discussion around leaf senescence would be helpful. Timing of leaf senescence could be an important indicator of whether leaves dropped due to stress during the growing season or if they were prompted to enter early dormancy.

You should indicate that SPAD readings were taken (and when). It would only take one sentence in the results to indicate no significant difference was found. However, the fact that there was no difference in SPAD and yet there were differences in leaf senescence warrants some discussion.

Reviewer 2 Report

Dear author and co-authors, dear editor,

The manuscript has been considerably improved by the conducted revisions. However, one remaining (minor) point is, that the discussion focuses mainly on below ground biomass and leaf senescence. The significant difference in above-ground biomass between O and M1 is still unmentioned.One critical flaw of the work, in general, is the lack of soil mineral N (SMN) analysis. Apparently, this can - unfortunately – not be revised afterwards. Particularly, when evaluating N fertilizer effects in pot experiments this is a crucial measure e.g. for assuring transferability of results to field conditions. If SMN is high (due to intensive soil disturbance, see L 59/60), additional fertilizer N has no or negative effects on plant growth. This should be critically discussed to avoid an overinterpretation of obtained results.
